# The Psychological Distress of Cancer Patients following the COVID-19 Pandemic First Lockdown: Results from a Large French Survey

**DOI:** 10.3390/cancers14071794

**Published:** 2022-03-31

**Authors:** Patricia Marino, Rajae Touzani, Jihane Pakradouni, Patrick Ben Soussan, Gwenaelle Gravis

**Affiliations:** 1Institut Paoli-Calmettes, SESSTIM, INSERM, IRD, Aix Marseille University, 13009 Marseille, France; rajae.touzani@inserm.fr; 2Department of Clinical Research and Innovation, Institut Paoli-Calmettes, 13009 Marseille, France; pakradounij@ipc.unicancer.fr; 3Department of Clinical Psychology, Institut Paoli-Calmettes, 13009 Marseille, France; bensoussanp@ipc.unicancer.fr; 4Department of Medical Oncology, Institut Paoli-Calmettes, Aix-Marseille University, CRCM, 13009 Marseille, France; gravisg@ipc.unicancer.fr

**Keywords:** COVID-19, cancer, post-traumatic stress disorder, anxiety

## Abstract

**Simple Summary:**

Cancer patients commonly experience anxiety which could increase with the COVID-19 pandemic situation. The aim of this study was to measure post-traumatic stress disorder (PTSD) and anxiety in cancer patients in France following the first COVID-19-related lockdown and associated factors. We found that the factors associated with PTSD and anxiety were different in nature. Factors associated with PTSD were not related to cancer but to the COVID-19 crisis, while factors associated with anxiety were mainly cancer related. More specifically, the fear of coming to hospital because of the risk of contracting COVID-19 was the strongest predictor of PTSD, and a better lockdown experience was protective against PTSD. Fear of cancer recurrence appear to be the main predictor of anxiety. Our study highlights the need to better integrate psychosocial support in pandemic response measures. Healthcare providers should not prioritize COVID-19 over cancer as the latter remains a central concern for cancer patients.

**Abstract:**

Cancer patients commonly experience psychological distress that may increase with the current COVID-19 pandemic. This prospective study aimed to measure post-traumatic stress disorder (PTSD) and anxiety in cancer patients following France’s first COVID-19-related lockdown, together with associated factors. Cancer patients receiving outpatient treatment or post-treatment follow-up completed a questionnaire which measured, among other things, PTSD (IES-R), anxiety (State-Trait Anxiety Inventory), and fear of cancer recurrence (FCR). Of the 1097 patients included in the study, 14.7% and 30.5% suffered from PTSD and anxiety, respectively. Patients afraid to come to hospital due to the risk of COVID-19 transmission (OR = 3.49, *p* < 0.001), those with a negative lockdown experience (OR = 0.98, *p* < 0.001), women (OR = 1.97; *p* = 0.009), and patients living alone (OR = 1.63, *p* = 0.045) were all more likely to have PTSD. Older patients (OR = 1.65, *p* = 0.020), women (OR = 1.62, *p* = 0.018), those with a higher FCR score (OR = 5.02, *p* < 0.001), patients unsatisfied with their cancer management (OR = 2.36, *p* < 0.001), and those afraid to come to hospital due to COVID-19 (OR = 2.43, *p* < 0.001) all had a higher risk of anxiety. These results provide a greater understanding of the psychological consequences of the COVID-19 pandemic in cancer patients and highlight the need to better integrate psychosocial support in pandemic response measures in order to guide health systems.

## 1. Introduction

Coronavirus Disease 2019 (COVID-19), which is caused by the SARS-CoV-2 virus, rapidly spread throughout the world in the early months of 2020. Faced with this large-scale health crisis, cancer care systems worldwide were restructured [1,2,3]. Specifically, remote consultations were put in place, certain surgical and medical procedures were postponed, anticancer treatment protocols were adjusted, enrolment in clinical trials was stopped, and restrictive measures were implemented in hospitals to control the flow of people. In France, the first national lockdown was implemented between 17 March and 13 May 2020 in order to contain viral circulation and limit hospital saturation, especially in intensive care units. 

At the beginning of the pandemic, the results of the first studies suggested that patients with cancer were exposed to a higher risk of infection with COVID-19 [4,5] and are more likely to experience poorer outcomes after contracting the disease with an increased risk of severe complications or mortality [6,7]. Thus, a recent large study found that patients with recent cancer treatment were at higher risk of hospitalization and death [8]. Managing cancer patients during the current pandemic has therefore been a particular challenge as minimizing this population’s exposure to SARS-CoV-2 has become a priority [9,10,11].

Cancer patients feel that they are a particularly vulnerable population with respect to COVID-19. Specifically, they have a high perceived risk of COVID-19 exposure and are afraid of developing acute symptoms [12,13]. These patients already experience very high levels of anxiety, depression and post-traumatic stress related to their cancer [14,15]. Accordingly, the current pandemic is a potential cause for increased anxiety arising from two factors: the risk of SARS-CoV-2 exposure if they come to hospital for cancer treatment, and the risk of poorer cancer outcomes—specifically the risk of cancer progression and death—due to COVID-19-related delays or changes in cancer treatment [16,17,18,19].

The objective of this study was to measure psychological distress (post-traumatic stress, anxiety) in cancer patients in France following the end of the country’s first COVID-19-related lockdown, and to identify associated factors including sociodemographic and medical characteristics, patient perception of their cancer management, their fear of cancer recurrence, and their lockdown experience.

## 2. Material and Methods

### 2.1. Participants

A cross-sectional self-rated survey was conducted in June 2020, the month following the end of France’s first national COVID-19-related lockdown, in a monocentric comprehensive cancer centre in Marseille. Patients were asked to participate by postal mail; they received an envelope which included an information letter, a questionnaire, and a pre-stamped envelope to return the completed questionnaire. Returning the questionnaire was considered as providing written informed consent to participate. Study approval was obtained from the institutional ethics committee (IRB # IPC 2020-026) and the French national ethics committee “Comité de Protection des Personnes Sud Méditerranée 5” (reference n°20.05.06.51657).

Eligible participants were adult patients with cancer (haematological or solid tumour) over 18 years of age receiving outpatient treatment or post-treatment follow-up.

### 2.2. Assessments Tools

The State-Trait Anxiety Inventory (STAI) questionnaire was used to assess anxiety [20]. This 40-item self-report scale assesses different dimensions of “state” (STAI-S) and ‘trait’ anxiety (STAI-T). We used for our analysis the subscale that measures the state of anxiety at the time of the study (20 items of STAI-S). The total anxiety score ranges from 20 to 80, with higher scores indicating greater anxiety.

Post-Traumatic Stress Disorder (PTSD) was assessed using the Impact of Event Scale-Revised (IES-R), which contains 22 items with Likert scale responses from 0 (not at all) to 4 (extremely) [21,22]. The total score ranges from 0 to 88, and higher total scores are suggestive of more severe PTSD symptoms. 

Fear of Cancer Recurrence (FCR) was assessed using the nine-item severity subscale of the Fear of Cancer Recurrence Inventory [23] where responses are rated on a Likert scale from 0 (not at all) to 4 (very much) with higher scores indicating higher levels of FCR.

Participants also answered six questions exploring different dimensions of their COVID-19 lockdown experience as follows: ability to stay busy, to relax, to manage stress, to see the positive aspects of the lockdown, to limit screen time, and to maintain family and social contacts. A summary score, ranging from 0 to 100, was constructed as a proxy of “lockdown experience”, with a higher score value indicating a better experience of the lockdown period. Other variables related to the COVID-19 health crisis were collected: living status during lockdown, patients’ perception of care during the pandemic (fear of COVID-19 infection, satisfaction with cancer care, satisfaction with measures to reduce exposure to COVID-19, perceived risk of COVID-19 infection, usefulness of phone consultations and psychological support need.

The questionnaire also contained questions on cancer-related variables (type of cancer, cancer management phase). Finally, socio-demographic characteristics and one question about having experienced difficult life events in the previous 6 months were also included.

### 2.3. Statistical Analysis

Categorical and quantitative variables were expressed as percentages, mean and standard deviation, respectively. In line with the literature, we chose to use a cut-off score of 33 to indicate PTSD diagnosis [22] and a cut-off of 45 for the STAI-S to define clinical anxiety [20]. In the univariate analysis, the influence of individual characteristics on anxiety level, PTSD, and fear of recurrence were analysed using Chi-squared tests (qualitative variables) or Student’s *t*-tests (quantitative variables). In the multivariate analyses, factors associated with anxiety and PTSD were analysed using an ordinal logistic regression model. A *p*-value less than 0.05 in the multivariate analyses was considered as the significance level. All the analyses were performed in the STATA software program, version 17.0 (StataCorp., College Station, TX, USA).

## 3. Results

### 3.1. Description of the Study Population

Of the 4000 French cancer patients contacted by postal mail, 1097 completed and returned the questionnaire. Their baseline characteristics are shown in Table 1. Participants were aged between 22 and 92 years old, 63.2% were female, and less than half had a third-level education level (48.8%). Most of the patients were over 50 years old (47.7% between 51 and 70 years old and 37.9% over 70 years old) and most did not live alone at home (75.0%). Before the first COVID-19-related lockdown, 60.1% were already retired and only 15.9% were still active. Just under half (41.4%) had breast/gynaecologic cancer, while 24.3% and 15.8% had hematologic and digestive cancer, respectively. Most participants reported they were receiving treatment or were in the follow-up phase for their cancer at the time of the study, while 8.7% were newly diagnosed. Almost two-thirds (64.9%) were afraid of cancer recurrence.

Regarding the factors related to the COVID-19 health crisis, a large proportion of respondents were not alone during the lockdown period (75.0%) and few participants reported contracting COVID-19 (7.7%). With respect to the lockdown period, 76.0% reported they managed to stay busy, 72.6% maintained family and/or social links, while only 21.9% and 34.4% were able to limit screen time and manage their stress, respectively (Figure 1). The six items presented in Figure 1 allowed us to calculate a summary score of the lockdown experience ranging from 0 to 100 for 893 respondents. In terms of hospital care, respondents were fully satisfied with the management of their disease during the health crisis (74.2%) and the measures put in place in hospital to reduce exposure to COVID-19 (76.0%). Most patients were not afraid to come to hospital for treatment during the pandemic (80.4%). The majority of patients did not seek consultations with a psychologist or psychiatrist (84.9%).

### 3.2. Univariate Analyses 

Univariate Analyses Carried out on Both Anxiety and PTSD Levels Are Presented in Table 1.

We found that 30.5% (*n* = 293) of patients suffered from anxiety and 14.7% suffered from PTSD (IES-R score ≥ 33) (Table 1).

COVID-19 variables associated with both anxiety and PTSD were (all *p* < 0.001): satisfaction with the current cancer management, satisfaction with the measures put in place at the hospital to reduce exposure to COVID-19, fear of coming to the hospital because of COVID-19 and the overall score of lockdown experience.

The only cancer-related variables associated with both anxiety and PTSD was the fear of cancer recurrence (*p* < 0.001).

Finally, two sociodemographic variables were significantly associated with both anxiety and PTSD: gender (*p* < 0.001) and the experience of difficult elements of life in the previous months (*p* < 0.001).

### 3.3. Factors Associated with Anxiety One Month Following COVID-19-Related Lockdown

Multivariate analysis highlighted that the likelihood of being anxious was greater in women (OR = 1.62, 95% CI = [1.09–2.41]), in patients over 70 years old (OR = 1.65; 95% CI = [1.08–2.51]), those about to start their treatment (OR = 2.31; 95% CI = [1.25–4.28]), and persons who had experienced difficult life events in the previous 6 months (OR = 2.11; 95% CI = [1.45–3.07]. In addition, patients who were not satisfied with the management of their cancer (OR = 2.36; 95% CI = [1.62–3.44]), those who were afraid to come to the hospital due to the risk of COVID-19 exposure (OR = 2.43; 95% CI = [1.61–3.67]) and those who were afraid of cancer recurrence (OR = 5.02; 95% CI = [3.07–8.18]); all had a greater risk of anxiety (Table 2).

### 3.4. Factors Associated with Post-Traumatic Stress Disorder One Month Following COVID-19-Related Lockdown

The multivariate analysis is presented in Table 2. We found that the probability of having PTSD symptoms was higher in patients who were afraid to come to the hospital because of the potential exposure to COVID-19 (OR = 3.49; 95% CI = [2.11–5.79]), in women (OR = 1.97; 95% CI = [1.18–3.29]), and in patients living alone at home (OR = 1.63; 95% CI = [1.01–2.63]). Furthermore, the probability of having PTSD decreased as the lockdown experience score increased (OR = 0.98; 95% CI= [0.97–0.99]).

## 4. Discussion

In this study, we aimed to assess the psychological impact (anxiety and PTSD) of France’s first COVID-19-related lockdown on cancer patients one month after the lockdown ended, and to analyse associated factors. We found that even one month after the end of the lockdown, PTSD and anxiety levels were high, with 14.7% and 30.5% of respondents suffering from PTSD and anxiety, respectively; these values reflect figures in several COVID-19 studies in the literature. In the review by Ayubi et al. [24], which assessed anxiety in studies that used various validated questionnaires, prevalence in cancer patients could exceed 30% during the pandemic. In the studies by JuanJuan et al. [25] and Joly et al. [18], moderate to severe PTSD symptoms (IES-R score ≥ 33) was found for 52.3% and 21% of patients, respectively. JuanJuan et al.’s study [25] had a particularly high prevalence of PTSD; this can be explained by the fact that it was conducted in the Hubei province of China—the epicentre of COVID-19—at the peak time point of the crisis in the country (February 2020). The prevalence in Joly et al.’s French study was higher than the value we found (21% vs. 14.7%), but that study was conducted during the first lockdown (April/May 2020), unlike ours, which was conducted a month after the lockdown ended. In any case, the prevalence of anxiety assessed after the COVID-19 pandemic were much higher than the 10% reported in a previous systematic review and meta-analysis conducted before the epidemic context [26,27].

One of the main results of our study is that the factors associated with PTSD and anxiety were different in nature. With regard to PTSD, factors were not related to cancer (cancer type, treatment phase, fear of cancer recurrence) but to the COVID-19 crisis. More specifically, the fear of coming to hospital for treatment because of the risk of contracting COVID-19 was the strongest factor associated with PTSD. Interestingly, our summary score describing patients’ lockdown experience was used as a proxy for the “quality” of their lockdown experience. We found that a higher summary lockdown score was associated with a decreased risk of PTSD, even after the lockdown ended. However, as this summary measure is not a validated tool, this result must be interpreted with caution and the effect of the lockdown experience on PTSD should be further investigated in future studies. To our knowledge this is the first study to assess the impact of lockdown experience on PTSD. Another study [28] found that maintaining good relationships with friends during the COVID pandemic was a protective factor against PTSD. Similarly, we also found that living alone was a predictor of PTSD. However, we found no association between cancer treatment phase and PTSD, which suggests that the stress caused by the COVID-19 pandemic affects all patients, even those who already have a good knowledge of the management of their cancer disease.

With regard to anxiety, associated factors were mainly cancer related, in particular the fear of cancer recurrence with a five-fold higher risk of anxiety for patients with a high FCR score. It is well known that the fear of cancer recurrence is highly associated with anxiety levels [29]. Our study highlights that in the COVID-19 era, cancer patients are most worried about the progression of their cancer. This finding highlights that healthcare providers should not prioritize COVID-19 over cancer as the latter remains a central concern for cancer patients.

Adjustments to oncology care delivery to reduce the risk of COVID-19, including delays in or disruption to treatment, expose patients to a very strong additional risk of anxiety [18,25,30]. This was confirmed in our study since dissatisfaction with cancer management during the COVID pandemic was a predictor for anxiety. In addition, just as we found for PTSD, the fear of coming to hospital for cancer treatment was a predictor of anxiety. In fact, the COVID-19 epidemic has emphasized anxiety in a vulnerable population already highly psychologically affected by cancer.

The physical and psychological side effects of cancer diagnosis, both during and after treatment, are well documented [31,32,33]. Studies indicate that emotional distress is at its most severe during diagnosis and active treatment [34]; however, clinical distress can persist in survivorship [35]. Accordingly, cancer patients are at high risk of developing clinical levels of emotional distress, which can result in affective disorders such as anxiety, depression and PTSD. The nature of the ongoing COVID-19 crisis may exacerbate this emotional distress symptomatology in this population.

Previous studies conducted in the general population have shown the negative impact of the COVID-19 pandemic on psychological health, including fear of illness, fear of death, as well as the social effects of physical distancing [36,37]. This stress caused by the COVID-19 outbreak may compound worrisome and ruminative thinking patterns that may predict worse outcomes for anxiety, depression and cognitive health in cancer patients, who already consistently indicate cognitive and emotional vulnerability. This suggests that the emotional stress experienced by cancer patients is over and above pre-existing levels because of COVID-19-related restrictions. Given that psychological distress in cancer patients has been associated with reduced treatment compliance, which may influence disease progression and mortality, any additional distress brought about by the COVID-19 pandemic is of great concern for this population.

Results of our study are in line with the study by Sigorski et al. [38], who showed, using a Numerical Anxiety Scale that COVID-19-related fear and anxiety were significantly lower than cancer-related anxiety. A large Chinese study [39] also showed that cancer patients who worried more frequently about their disease management due to COVID-19 were at a higher risk of mental health problems. In our study, the prevalence of fear of cancer recurrence was more than twice as high as that of anxiety. Our results also highlight that particular attention should be paid to newly diagnosed cancer patients who initiated a new treatment during the pandemic, as we found they were at very high risk of anxiety. Finally, in line with previous literature—whether in the COVID-19 context or not—women in our study were more likely to have anxiety and PTSD [18,28]. One should notice that the fact that women’s psychological health was more affected in our study was not related to the type of cancer as long as the multivariate analysis was adjusted for cancer location.

Our study provides additional knowledge to the existing literature and has several strengths. First, in most studies evaluating the consequences of the COVID-19 pandemic on the psychological health of cancer patients, patients were interviewed during a lockdown. At the time our survey was launched, the first French lockdown had just ended, and we were interested in evaluating psychologic health post lockdown. Second, while selection bias cannot be excluded, the generalizability of our results is greater than that of previous work, as our study population was not pre-selected. Third, the survey was conducted in patients with various cancer types, including haematological cancers for which data in the literature are limited. Fourth, it concerned day-hospital consultations, diagnosis disclosure appointments, and post-treatment follow-up consultations. Fifth, participants could be at any stage of cancer management (i.e., about to start treatment, on treatment, or during post-therapeutic follow-up). Finally, our sample size was larger than most other studies exploring the same topic.

The study also has limitations. First, its cross-sectional design did not allow us to establish any cause-and-effect relationships. Second, we did not have a control group (i.e., a pre-COVID-19 pandemic group) and were unable to assess patients’ initial state before COVID-19 pandemic occurred. Finally, the study was monocentric, which limits extrapolating our results to other settings.

In terms of our cancer institute, Institut Paoli Calmettes (Marseille, France), the onset of the COVID-19 crisis in 2020 led to a general reorganization of patient care, including visiting restrictions, appointment changes, and increased telehealth. Telephone-based psychological support was also proposed to patients and families. Our institute focused on remaining COVID-19-free, and all patients with the disease were referred to specialized COVID-19 hospitals. It is therefore likely that high rates of anxiety and PTSD we observed in our study population are underestimated compared to the rates one would observe in a sick population treated in a hospital receiving patients with COVID-19.

## 5. Conclusions

The ongoing COVID-19 pandemic continues to affect social life and health management. Researching the factors associated with COVID-19-related exacerbation of anxiety and post-traumatic stress disorder in cancer patients could lead to improved screening of these disorders in this population. Our findings highlight the need for efforts to better integrate psychosocial support into evolving pandemic response measures in order to guide health systems towards person-centred management. Although the COVID-19 response is ongoing and contexts are constantly evolving, how well we respond is ultimately dependent on how well we efficiently translate lessons learned into effective policy and practice.

## Figures and Tables

**Figure 1 cancers-14-01794-f001:**
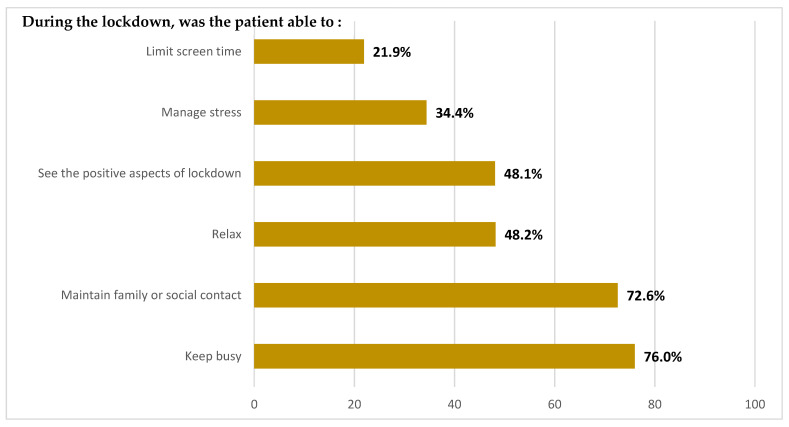
The different positive declarative experiences of lockdown (*n* = 893).

**Table 1 cancers-14-01794-t001:** Characteristics of the participants and factors associated with anxiety level and post-traumatic stress disorder during the COVID-19 health crisis: univariate analyses.

Variables	Total *n* = 1097	Anxiety Level (STAI-S) *n* = 962		Post-Traumatic Stress Disorder (IES-R) *n* = 810	
	*n* (%)	Absence of Anxiety ^+^ *n* = 669 (69.5%)	Presence of Anxiety ^++^ *n* = 293 (30.5%)	Low * Level of Post-Traumatic Stress Disorder *n* = 691 (85.3%)	Moderate/Severe ** Level of Post-Traumatic Stress Disorder *n* = 119 (14.7%)
	**Factors related of COVID-19 health crisis**				
**Contracted COVID-19**		*p* = 0.675		*p* = 0.100	
No	999 (92.3)	613 (92.5)	264 (91.7)	638 (93.1)	103 (88.8)
Yes	83 (7.7)	50 (7.5)	24 (8.3)	47 (6.9)	13 (11.2)
**Living status during lockdown**		*p* = 0.235		*p* = 0.137	
Living alone	228 (21.3)	126 (19.2)	67 (23.4)	138 (20.3)	31 (26.5)
Living with others	800 (75.0)	511 (77.9)	209 (72.8)	522 (76.6)	80 (68.4)
Was not in lockdown	39 (3.7)	19 (2.9)	11 (3.8)	21 (3.1)	6 (5.1)
**Overall ‘lockdown experience’ score** *Mean (SD)*		** *p* ** **< 0.001**		** *p* ** **< 0.001**	
	46.5 (26.1)	51.4 (25.9)	38.9 (24.8)	50.0 (26.6)	37.0 (24.2)
**Completely satisfied with the current management of their cancer**		** *p* ** **< 0.001**		** *p* ** **= 0.001**	
Yes	798 (74.2)	529 (80.1)	178 (61.8)	522 (76.3)	72 (62.1)
No	278 (25.8)	131 (19.9)	110 (38.2)	162 (23.7)	44 (37.9)
**Completely satisfied with the measures put in place at the hospital to reduce their exposure to COVID-19**		** *p* ** **< 0.001**		** *p* ** **= 0.001**	
Yes	769 (76.0)	497 (79.8)	189 (68.7)	510 (77.9)	72 (63.7)
No	243 (24.0)	126 (20.2)	86 (31.3)	145 (22.1)	41 (36.3)
**Fear of going to hospital for treatment because of the risk of COVID-19 contamination**		** *p* ** **< 0.001**		** *p* ** **< 0.001**	
No	858 (80.4)	555 (84.9)	204 (70.8)	582 (86.0)	71 (60.7)
Yes	209 (19.6)	99 (15.1)	84 (29.2)	95 (14.0)	46 (39.3)
**Consulted a psychologist or psychiatrist during the pandemic**		** *p* ** **< 0.001**		** *p* ** **= 0.001**	
No, never	910 (84.9)	573 (87.3)	231 (79.4)	586 (86.1)	92 (78.0)
Yes, only once	86 (8.0)	55 (8.4)	23 (7.9)	60 (8.8)	8 (6.8)
Yes, several times	76 (7.1)	28 (4.3)	37 (12.7)	35 (5.1)	18 (15.2)
	**Cancer-related factors**				
**Type of cancer**		** *p* ** **= 0.001**		*p* = 0.066	
Hematological	252 (24.3)	168 (26.5)	57 (20.2)	163 (24.7)	27 (23.5)
Urological	111 (10.7)	83 (13.1)	20 (7.1)	83 (12.6)	5 (4.3)
Digestive	164 (15.8)	97 (15.3)	43 (15.2)	104 (15.8)	16 (13.9)
Breast/Gynecological	429 (41.4)	232 (36.7)	142 (50.4)	259 (39.3)	57 (49.6)
Other	81 (7.8)	53 (8.4)	20 (7.1)	50 (7.6)	10 (8.7)
**Treatment phase**		** *p* ** **< 0.001**		*p* = 0.209	
On treatment	647 (60.6)	386 (59.4)	186 (64.1)	419 (62.1)	66 (56.4)
Post-treatment follow-up	327 (30.7)	223 (34.3)	68 (23.5)	206 (30.5)	37 (31.6)
About to initiate treatment	93 (8.7)	41 (6.3)	36 (12.4)	50 (7.4)	14 (12.0)
**Fear of cancer recurrence (severity subscale)**		** *p* ** **< 0.001**		** *p* ** **< 0.001**	
No (score < 13)	301 (35.1)	254 (44.3)	23 (10.0)	243 (41.0)	12 (11.8)
Yes (score ≥ 13)	557 (64.9)	319 (55.7)	206 (90.0)	349 (59.0)	90 (88.2)
	**Sociodemographic factors**				
**Gender**		** *p* ** **< 0.001**		** *p* ** **< 0.001**	
Women	691 (63.2)	376 (56.3)	220 (75.3)	406 (59.0)	91 (76.5)
Men	403 (36.8)	292 (43.7)	72 (24.7)	282 (41.0)	28 (23.5)
**Age (years)**		** *p* ** **= 0.002**		*p* = 0.074	
22–50	156 (14.4)	86 (13.0)	63 (21.6)	105 (15.4)	28 (23.9)
51–70	516 (47.7)	343 (52.0)	127 (43.6)	355 (52.1)	55 (47.0)
71–92	409 (37.9)	230 (35.0)	101 (34.7)	221 (32.5)	34 (29.1)
**Living alone at home**		*p* = 0.152		** *p* ** **= 0.012**	
No	806 (75.0)	506 (76.9)	206 (72.5)	522 (76.9)	78 (66.1)
Yes	269 (25.0)	152 (23.1)	78 (27.5)	157 (23.1)	40 (33.9)
**Education level**		*p* = 0.071		*p* = 0.282	
No schooling/primary school	95 (8.8)	44 (6.7)	22 (7.6)	44 (6.5)	12 (10.2)
Secondary school	457 (42.4)	264 (40.1)	137 (47.2)	298 (43.8)	46 (39.0)
Higher education	521 (48.8)	351 (53.2)	131 (45.2)	339 (49.8)	60 (50.8)
**Professional situation before lockdown**		** *p* ** **= 0.025**		** *p* ** **= 0.031**	
Active	173 (15.9)	112 (16.8)	47 (16.2)	122 (17.8)	22 (18.6)
Inactive	262 (24.0)	156 (23.5)	92 (31.7)	170 (24.8)	42 (35.6)
Retirement	654 (60.1)	397 (59.7)	151 (52.1)	394 (57.4)	54 (45.8)
**Had experienced difficult elements of life in the previous 6 months (i.e., before COVID-19 pandemic)**		** *p* ** **< 0.001**		** *p* ** **< 0.001**	
No	501 (45.7)	358 (53.6)	85 (29.1)	336 (48.7)	34 (28.6)
Yes	594 (54.3)	310 (46.4)	207 (70.9)	354 (51.3)	85 (71.4)

* low IES-R score < 33; ** IES-R score ≥ 33; **^+^** STAI Anxiety score ≤ 45; ^++^ STAI Anxiety score > 45.

**Table 2 cancers-14-01794-t002:** Factors associated with anxiety level and post-traumatic stress disorder during the COVID-19 health crisis: multivariate analyses.

Ref: Absence of Anxiety ^+^	Presence of Anxiety (STAI-S) ++ *n* = 763		Ref: No/Low-Level of Post-Traumatic Stress Disorder *	Moderate/Severe Level of Post-Traumatic Stress Disorder (IES-R) *n*= 635	
	aOR	95% CI		aOR	95% CI
**Gender** *(Ref: Men)*			**Gender** *(Ref: Men)*		
Woman	1.62	1.09–2.41	Woman	1.97	1.18–3.29
**Age** *(Ref: 51–70)*			**Living alone at home** *(Ref: No)*		
22–50	1.54	0.97–2.44	Yes	1.63	1.01–2.63
71–92	1.65	1.08–2.51	**Overall ‘lockdown experience’ score**	0.98	0.97–0.99
**Treatment phase** *(Ref: On treatment)*	0.83	0.55–1.25	**Fear of coming to the hospital for treatment because of the risk of COVID-19 contamination** *(Ref: No)*		
Post-treatment follow-up			Yes	3.49	2.11–5.79
About to initiate treatment	2.31	1.25–4.28			
**Fear of cancer recurrence (severity subscale)** *(Ref:* No (score < 13))					
Yes (score ≥ 13)	5.02	3.07–8.18			
**Had experienced difficult elements of life in the previous 6 months (i.e., before COVID-19 pandemic)** *(Ref: No)*					
Yes	2.11	1.45–3.07			
**Completely satisfied with the current management of their cancer** *(Ref: Yes)*					
No	2.36	1.62–3.44			
**Fear of going to hospital for treatment because of the risk of COVID-19 contamination** *(Ref: No)*					
Yes	2.43	1.61–3.67			

Ref.: reference category; aOR: adjusted odds ratio; 95% CI: 95% confidence interval; ^+^ STAI Anxiety score ≤ 45; ^++^ STAI Anxiety score > 45; * IES-R score < 33.

## Data Availability

The data presented in this study are available on request from the corresponding author.

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
