# Peer review of "The Psychological Distress of Cancer Patients following the COVID-19 Pandemic First Lockdown: Results from a Large French Survey"

_cancers, 2022, doi:10.3390/cancers14071794_

Round 1
Reviewer 1 Report
This well-written paper deals with a relevant topic: the impact of the COVID-19 pandemic following France’s first lockdown on the psychological health of cancer patients. This article may be appropriate for publication with major revisions of the following points:
- (line 51) "and are more likely to have worse outcomes after contracting the disease". what are the worst outcomes? It may be helpful to be more specific.
- (line 82-85) it is not clear what type of STAI was used? (in table 2 reference is made to the STAI-AE): specify and specify the cut off used
- (line 110) The cut-off used for the IESS is very large (>33), so there will certainly be an overestimation of the results and this has a significant impact in terms of results and clinical impact. In the literature it is known that the cut off>37 discriminates the presence of PTSD symptoms: I would consider it more appropriate to keep the most reliable value to discriminate the probable presence of the symptoms (remember that from a questionnaire diagnosis is made, so it is always advisable to use the cut-offs that can best explain the presence of a certain phenomenon). In the explanation of the IESS-R tool, I would also include the following cut offs: 33–36 (moderate psychological impact) and> 37 (severe psychological impact)
- (line 139) the average score of six questions of a non validated tool doesn't make much sense, so I ask you to remove the average score. Also in the discussion be careful to draw too strong conclusions from these items (which can be discussed in a qualitative way)
- (line 142-143) "Not surprisingly" .. "Despite the potentially anxious context": remove comments and judgments in the presentation of the results (these sentences can be reported in the chapters relating to the discussion of the data).
- It is superfluous to insert also the table 2 of the univariate analyzes, as this analysis led to the multivariate analysis, which takes into account the influence of the multiple variables considered. Presenting the data in Table 2 may lead to a partial and speculative interpretation of the data. It is more correct to remove table 2 and the text part (delete 149 to 194) and insert the results of table 3 into the text.
- Review the discussions with what has been said so far, and remove in the part relating to the data that emerged in table 2;
- (lines 221 and 222) is a very strong statement, since the score is not of a validated instrument (it can mean everything and nothing and lead to interpretative biases): remove this statement as it is very questionable, or rewrite it underlining that "it would seem that ... "and" this aspect should be investigated further ... "
Author Response
REVIEWER 1
This well-written paper deals with a relevant topic: the impact of the COVID-19 pandemic following France’s first lockdown on the psychological health of cancer patients. This article may be appropriate for publication with major revisions of the following points:
- (line 51) "and are more likely to have worse outcomes after contracting the disease". what are the worst outcomes? It may be helpful to be more specific.
Authors’ response: Thank you very much for your suggestion. The following sentence was added to answer the reviewer’s comment (line 52): “… with an increased risk of severe complications or mortality [6,7]. Thus a recent large study found that patients with recent cancer treatment were at higher risk of hospitalization and death [8]
The following reference was added:
[8] MC Mac Gregor, X Lei, Zhao H et al. Evaluation of COVID-19 Mortality and Adverse Outcomes in US Patients With or Without Cancer. Jama Oncol 2022 ;8(1):69-78. doi:10.1001/jamaoncol.2021.5148
- (line 82-85) it is not clear what type of STAI was used? (in table 2 reference is made to the STAI-AE): specify and specify the cut off used
Authors’ response: in our analyses, we used the subscale in the questionnaire that evaluates the anxiety-state (Anxiété Etat in French-AE) which reflects the current anxiety State (STAI-S). It makes it possible to evaluate the nervousness and worry of the patient at time T.
To clarify this point, in the assessment tools section of the article:
- we completed the sentence describing the STAI questionnaire: “This 40-item self-report scale assesses different dimensions of ‘state’ (STAI-S) and ‘trait’ anxiety (STAI-T)”.
- we added the following sentence: “We used for our analysis the subscale that measures the state of anxiety at the time of the study (20 items of STAI-S).”
We also have replaced in the tables STAI-AE by STAI-S.
The cut off used for the STAI anxiety scale is 45, based on the original article by Spielberger et al [20]. For greater clarity, we have completed the sentence which specify the cut-off in the "Statistical analysis" section: “…and a cut-off of 45 for the STAI-S to define clinical anxiety [20]”.
- (line 110) The cut-off used for the IESS is very large (>33), so there will certainly be an overestimation of the results and this has a significant impact in terms of results and clinical impact. In the literature it is known that the cut off>37 discriminates the presence of PTSD symptoms: I would consider it more appropriate to keep the most reliable value to discriminate the probable presence of the symptoms (remember that from a questionnaire diagnosis is made, so it is always advisable to use the cut-offs that can best explain the presence of a certain phenomenon). In the explanation of the IESS-R tool, I would also include the following cut offs: 33–36 (moderate psychological impact) and> 37 (severe psychological impact)
Authors’ response: Thank you for your comment. Different cut-off has been suggested for IES-R scale, ranging from 22 (Rash et al, Addict Behav 2008 Aug;33(8):1039-47) to 44 (Blake et al. J Trauma Stress 1995 Jan;8(1):75-90).
Our choice of a cut-off >33 relied on the article by Creamer et al. [22] "Psychometric properties of the Impact of Event Scale—Revised". In this article, the authors specified that "a cutoff of 1.5 (equivalent to a total score of 33) was found to provide the best diagnostic accuracy." According to the authors, above this value, patients are considered to have clinically significant PTSD [22]. Besides, this cut-off of 33 is used in various studies analyzing PTSD in cancer population (ref [18] in our article for example).
- (line 139) the average score of six questions of a non validated tool doesn't make much sense, so I ask you to remove the average score. Also in the discussion be careful to draw too strong conclusions from these items (which can be discussed in a qualitative way)
Authors’ response: In agreement with reviewer comment we deleted the average score of the 6-items relative to the experience of lockdown in the text. This average score was also deleted from Figure 1.
Moreover, as suggested, the conclusion relative to this score was put into perspective in the Discussion Section (cf also suggestion below by the reviewer). We therefore modified the sentence as follows “We found that a higher summary lockdown score was associated with a decreased risk of PTSD. However, as this summary measure is not a validated tool, this result must be interpreted with caution and this aspect should be investigated in further studies” (line 216).
- (line 142-143) "Not surprisingly" .. "Despite the potentially anxious context": remove comments and judgments in the presentation of the results (these sentences can be reported in the chapters relating to the discussion of the data).
Authors’ response: We agree with the remark and comments and judgements were deleted in the “Results” Section.
- It is superfluous to insert also the table 2 of the univariate analyzes, as this analysis led to the multivariate analysis, which takes into account the influence of the multiple variables considered. Presenting the data in Table 2 may lead to a partial and speculative interpretation of the data. It is more correct to remove table 2 and the text part (delete 149 to 194) and insert the results of table 3 into the text.
Authors’ response:
We had initially chosen to present the univariate analyses with the objective of showing all the variables considered in our analyses that could have been included in the multivariate model.
Since all the available variables are now listed in Table 1, we believe it may be appropriate to include the univariate analysis in Table 1 as part of the descriptive analysis. So we merged our tables 1 and 2.
The text has been modified accordingly with the univariate analyses presented in a separate paragraph (part 3.2) with a shortened text.
Multivariate analyses are presented in parts 3.3 and 3.4 with added odds ratio in the text.
According reviewer comments we hope these modifications are appropriate.
- Review the discussions with what has been said so far, and remove in the part relating to the data that emerged in table 2;
The results that are included and discussed in the “Discussion Section” refer to the univariate analyses, as suggested by the reviewer.
- (lines 221 and 222) is a very strong statement, since the score is not of a validated instrument (it can mean everything and nothing and lead to interpretative biases): remove this statement as it is very questionable, or rewrite it underlining that "it would seem that ... "and" this aspect should be investigated further ... "
In accordance with the reviewer’s comment, we modified the sentence as follows: “We found that a higher summary lockdown score was associated with a decreased risk of PTSD. However, as this summary measure is not validated, this result must be interpreted with caution and the effect of the lockdown experience on PTSD should be further investigated in future studies”. (line 216)
Reviewer 2 Report
Title and Abstract: you are talking about a prospective study, but as I understand, your survey was all handed to the patients in one envelope with no prospective assessment. Thus, you should not talk about a prospective assessment in the title or abstract to not mislead the reader. In the limitation you rightly state that your design is cross-sectional.
Line 50: I would only cite study 4 (and not 5) here and tone down the sentence. Study 4 did not demonstrate but assumed that cancer patients are more likely to become infected then the other patients. [“18 (1%; 95% CI 0.61–1.65) of 1590 COVID-19 cases had a history of cancer, which seems to be higher]. In fact the CI included 1 and was not significant. The main point of the studies is that under specific circumstances cancer patients have an increased mortality risk or poorer prognosis.
Line 63: Your title implies that the focus of the study is on the negative impact of the pandemic to the mental health of cancer patients. Within your objectives, you assess more pandemic unrelated factors (sociodemographic, medical, fear of cancer recurrence) than specific pandemic related factors (lockdown experience, and perception of cancer management during the lockdown – which should be specified). I would suggest that you either adjust the title to your objectives or reformulate the objectives and focus on the specific pandemic related factors for the main analyses. Also it is useful (as a replication of past research) to verify that the other sociodemographic factors like age and gender are equally affecting mental health like in other populations. That helps to generalize your results.
Line 55: Please provide a reference for the statement you make here about the perception of cancer patients of being vulnerable and at risk.
Linen 82: Since you want to analyses the impact of the pandemic on mental health outcomes, I wonder why you specifically selected an anxiety questionnaire that differentiates between state and trait anxiety (thereby including individual personality). This could confound the results regarding the general impact of the pandemic during a specified period. Or did you have other questionnaires in the survey you can rely on?
Line 100 ff: In the previous paragraph you describe your main variable of interest (according to the focus you put in your title) which is the lockdown experience. From line 100 onwards you mix again pandemic related factors (perception of care during the pandemic, infection status) with unrelated factors (e.g. sociodemographics). For streamlining your arguments I would restructure the reporting of variables and differentiate between pandemic and non-pandemic predictors for mental health.
Line 102: Did you also assess how often caring has taken place during the study period? Number of phone calls, etc.)
Line 110: Rethink formulation: “Persons with an IES-R total score over 33 were considered to have PTSD symptoms”…are the items not generally measuring symptoms? Do you mean that patients with a score above 33 are supposed to have a clinically relevant PTSD?
Figure 1: There is a grammatical error in the question formulation: “During the lockdown, did the patient was able to:”Either did or was able to
Line 171: here the reference for the multivariate analyses should be table 3 not 2.
Table 2 & 3: What does AE mean behind STAI?
Table 2: reconsider wording of the last two columns. As it reads now (low vs. moderate/severe) it is suggesting that all individuals have some type of PTSD, even though you specify the value in the note section. Maybe consult the manual again if it would be ok to talk about no PTSD?
Table 2: Why did you not report on the satisfaction with cancer management during the pandemic and only report on the overall satisfaction? The question is also differently formulate in table 1 and 2 (totally, vs. completely).
In this sense, it would also help the reader if you maintain the wording for each factor throughout the paper (in the text and tables). For example, in the table you talk about cancer-related factors, in the text about medical conditions. The good thing is that you clearly differentiate in the table the pandemic factors from the other factors. Question would be if you put the focus on the pandemic factors and put them in first in the table and not last and talk about main analyses and additional analyses?
Discussion:
Line 200: to point out to the need for cancer patient care during a pandemic it would be interesting to compare prevalence ratings of cancer patients in studies conducted before the pandemic and not so much comparing them with similar studies during the pandemic. If your prevalence rates are higher than they are “normally” reported, it would strengthen your argument for special care of cancer patients.
Line 215: According to my previous comments I would reformulated your statement, since your main goal was to see, what impact pandemic related factors have on patient mental health…
Line 2018: Since your study is cross-sectional in nature, because you assessed the variables of interest within one survey, I would use less strong words like “predictor” of PTSD. PTSD could also be a predictor of increased fear.
Line 219: this information about the summary score is for the method section and should not be repeated in the discussion.
Line 231: Please do not repeat the results in the discussion section. Especially regarding this paragraph, a comparison of the prevalence rates of cancer patients in general would be helpful. Maybe you can also compare whether the fear of recurrence is higher or equal to previous studies with cancer patients before the pandemic.
Line 238: I see a discrepancy between the question you raised about satisfaction with cancer treatment and the specific interpretation you make regarding disruption to treatment or delays. If these aspects were not specifically asked in the question, it is difficult to know, which treatment aspects patients did consider. Maybe they were not satisfied with a physician or unfriendly staff etc. As it reads now, the causes for dissatisfaction cannot be identified.
Line 250-251: see my comments above. If you would compare prevalence rates of PTSD and anxiety of cancer patients before the pandemic you could make a stronger statement here.
Line 252: Again I would tone down the implications. You did not explicitly measure additive stress (you measured positive lockdown experiences). And while fear of coming to the hospital due to the risk of a contamination is significant in both multivariate analyses, the lockdown experience is only significant for PTSD. Regarding anxiety, the strongest effect was found for fear of recurrence.
Line 264: If you want to make this argument you should more explicitly state the prevalence for fear of recurrence of 64.9% which is more than twice as high as the anxiety score of 30,5% for your study (without naming the exact numbers in the discussion).
Line 269-270: Please avoid one sentence paragraphs. This information is a replication of past studies and is good to mention, since it strengthens the generalizability of your results. But, I would add some information on that and maybe other replicated regarding demographics or medical issues. That women are more affected also fits to the result that gynecological cancer is more relevant in anxiety than urological cancer.
Line 286: here also a comparative prevalence rate of cancer patient’s mental health regarding anxiety and PTSD would help.
Author Response
REVIEWER 2
Title and Abstract: you are talking about a prospective study, but as I understand, your survey was all handed to the patients in one envelope with no prospective assessment. Thus, you should not talk about a prospective assessment in the title or abstract to not mislead the reader. In the limitation you rightly state that your design is cross-sectional.
Authors’ response: We deleted the word “prospective” in the title to not mislead the reader.
Line 50: I would only cite study 4 (and not 5) here and tone down the sentence. Study 4 did not demonstrate but assumed that cancer patients are more likely to become infected then the other patients. [“18 (1%; 95% CI 0.61–1.65) of 1590 COVID-19 cases had a history of cancer, which seems to be higher]. In fact the CI included 1 and was not significant. The main point of the studies is that under specific circumstances cancer patients have an increased mortality risk or poorer prognosis.
Authors’ response: We must acknowledge that when this study was conducted, the results of the first scientific studies concluded that there was a link between cancer and COVID-19 infection. In response to the reviewer comment, we modified the sentence as follows: “At the beginning of the pandemic, the results of the first studies suggested that patients with cancer were exposed to a higher risk of infection with COVID-19”. (line 50)
As requested by the reviewer, we deleted reference 5, and we have replaced the reference with this one:
[5] Dai M, Liu D, Zhou F and al. Patients with cancer appear more vulnerable to SARS-CoV-2: a multicenter study during the COVID-19 outbreak. 2020 Cancer Discov, 10 (June (6)), pp. 783-791
Line 63: Your title implies that the focus of the study is on the negative impact of the pandemic to the mental health of cancer patients. Within your objectives, you assess more pandemic unrelated factors (sociodemographic, medical, fear of cancer recurrence) than specific pandemic related factors (lockdown experience, and perception of cancer management during the lockdown – which should be specified). I would suggest that you either adjust the title to your objectives or reformulate the objectives and focus on the specific pandemic related factors for the main analyses. Also it is useful (as a replication of past research) to verify that the other sociodemographic factors like age and gender are equally affecting mental health like in other populations. That helps to generalize your results.
Authors’ response: We agree with the reviewer and the title was therefore modified: “The psychological distress of cancer patients following the COVID-19 pandemic first lockdown: results from a large French survey.”
Line 55: Please provide a reference for the statement you make here about the perception of cancer patients of being vulnerable and at risk.
Authors’ response: We agree with the reviewer; a reference is lacking. Therefore, we added the 2 following references:
[12] Miaskowski, C, Paul, SM, Snowberg, K, Abbott, M, Borno, H, Chang, S, et al. Oncology patients’ perceptions of and experiences with COVID-19. Support Care Cancer. 2021;29(4):1941-50
[13] Guven DC, Sahin TK, Aktepe OH, Yildirim HC, Aksoy S, et al. Perspectives, knowledge, and fears of cancer patients about COVID-19. Front Oncol. 2020;10:1553–1553. doi: 10.3389/fonc.2020.01553.
Linen 82: Since you want to analyses the impact of the pandemic on mental health outcomes, I wonder why you specifically selected an anxiety questionnaire that differentiates between state and trait anxiety (thereby including individual personality). This could confound the results regarding the general impact of the pandemic during a specified period. Or did you have other questionnaires in the survey you can rely on?
Authors’ response: We decided to include this questionnaire because we were in a cross-sectional design in the absence of a baseline value. We do not have another measure of anxiety in the questionnaire.
Line 100 ff: In the previous paragraph you describe your main variable of interest (according to the focus you put in your title) which is the lockdown experience. From line 100 onwards you mix again pandemic related factors (perception of care during the pandemic, infection status) with unrelated factors (e.g. sociodemographics). For streamlining your arguments I would restructure the reporting of variables and differentiate between pandemic and non-pandemic predictors for mental health.
Authors’ response: The text of the article was modified in accordance with the reviewer’s comment in order to differentiate between pandemic and non-pandemic factors associated with mental health:
“Other variables related to the COVID-19 health crisis were collected: living status during lockdown, patients’ perception of care during the pandemic (fear of COVID-19 infection, satisfaction with cancer care, satisfaction with measures to reduce exposure to COVID-19, perceived risk of COVID-19 infection usefulness of phone consultations, psychological support need”. (line 103)
“The questionnaire also contained questions: on cancer-related variables (type of cancer, cancer management phase). Finally, socio-demographic characteristics and one question about having experienced difficult life events in the previous 6 months were also included.” (line 108)
Line 102: Did you also assess how often caring has taken place during the study period? Number of phone calls, etc.)
Authors’ response: No this information was not collected in the questionnaire.
Line 110: Rethink formulation: “Persons with an IES-R total score over 33 were considered to have PTSD symptoms”…are the items not generally measuring symptoms? Do you mean that patients with a score above 33 are supposed to have a clinically relevant PTSD?
Authors’ response: Our choice of a cut-off >33 relied on the article by Creamer et al. [22] "Psychometric properties of the Impact of Event Scale—Revised". In this article, the authors specified that "a cutoff of 1.5 (equivalent to a total score of 33) was found to provide the best diagnostic accuracy." According to the authors, above this value, patients are considered to have clinically significant PTSD [22]. Besides, this cut-off of 33 is used in various studies analyzing PTSD in cancer population.
Figure 1: There is a grammatical error in the question formulation: “During the lockdown, did the patient was able to:”Either did or was able to
Authors’ response: Thank you for this comment. The title of figure 1 was modified as follows “During the lockdown, was the patient able to:”
Line 171: here the reference for the multivariate analyses should be table 3 not 2.
Authors’ response: The mistake was modified in the text.
Table 2 & 3: What does AE mean behind STAI?
Authors’ response: there is an error of translation. The STAIT-AE (Anxiété Etat in French) is the subscale of the anxiety state (STAI-S in English).
We therefore have replaced in the tables STAI-AE by STAI-S (which is now described in the “Methods” Section, lines 85-87).
Table 2: reconsider wording of the last two columns. As it reads now (low vs. moderate/severe) it is suggesting that all individuals have some type of PTSD, even though you specify the value in the note section. Maybe consult the manual again if it would be ok to talk about no PTSD?
Authors’ response: The reviewer is right. The wording in the column was modified in “no/low PTSD”
Table 2: Why did you not report on the satisfaction with cancer management during the pandemic and only report on the overall satisfaction?
Authors’ response: The time of analysis was the end of the lockdown period in France, the treatment restrictions were still in place at the time when the patients completed the questionnaire. Therefore, patients were asked about how their disease was being managed at the time of the study (it means during the COVID-19 restriction period).
The question is also differently formulate in table 1 and 2 (totally, vs. completely).
Authors’ response: According reviewer’s comment, the wording of the questions has been harmonized in the two tables.
In this sense, it would also help the reader if you maintain the wording for each factor throughout the paper (in the text and tables). For example, in the table you talk about cancer-related factors, in the text about medical conditions. The good thing is that you clearly differentiate in the table the pandemic factors from the other factors. Question would be if you put the focus on the pandemic factors and put them in first in the table and not last and talk about main analyses and additional analyses?
Authors’ response: In accordance with the reviewer’s comment, we harmonized the wording in text and tables.
Moreover, the factors associated with COVID-19 were put in first in the tables, as suggested.
Discussion:
Line 200: to point out to the need for cancer patient care during a pandemic it would be interesting to compare prevalence ratings of cancer patients in studies conducted before the pandemic and not so much comparing them with similar studies during the pandemic. If your prevalence rates are higher than they are “normally” reported, it would strengthen your argument for special care of cancer patients.
Authors’ response: A systemic review and meta-analysis showed that the prevalence rate of anxiety was 10% in patients treated for cancer.
As suggested by the reviewer, we now mention the prevalence of anxiety in cancer patients outside the epidemic context in the following sentence: In any case, the prevalence of anxiety assessed after the COVID-19 pandemic were much higher than the 10% reported in a previous systematic meta-analysis and a recent study, conducted before the epidemic context [26,27].
[26] Mitchell A, Chan M, Bhatti H, et al. Prevalence of depression, anxiety, and adjustment disorder in oncological, haematological, and palliative-care settings: a meta-analysis of 94 interview-based studies. Lancet Oncol. 2011 Feb;12(2):160-74. doi: 10.1016/S1470-2045(11)70002-X.
[27] Pitman A, Suleman S, Hyde N, Hodgkiss A. Depression and anxiety in patients with cancer. BMJ. 2018;361:k1415. doi: 10.1136/bmj.k1415
Line 215: According to my previous comments I would reformulated your statement, since your main goal was to see, what impact pandemic related factors have on patient mental health…
Authors’ response: The has been reworded to fit our objective that was to look at the psychological health (anxiety and PTSD) of the patients after the 1st confinement, and to find the factors associated, notably those related to the epidemic context.
Line 2018: Since your study is cross-sectional in nature, because you assessed the variables of interest within one survey, I would use less strong words like “predictor” of PTSD. PTSD could also be a predictor of increased fear.
Authors’ response: We agree with the reviewer and we therefore modified the sentence as follows: “More specifically, the fear of coming to hospital for treatment because of the risk of contracting COVID-19 was the stronger factor associated with PTSD”. (line 213)
Line 219: this information about the summary score is for the method section and should not be repeated in the discussion.
Authors’ response: In accordance with the reviewer comment, we have deleted a part of the sentence which referred to the method. (lines 213-216).
Line 231: Please do not repeat the results in the discussion section. Especially regarding this paragraph, a comparison of the prevalence rates of cancer patients in general would be helpful. Maybe you can also compare whether the fear of recurrence is higher or equal to previous studies with cancer patients before the pandemic.
Authors’ response: As requested by the reviewer the sentence was modified “… in particular the fear of recurrence with a five-fold higher risk of anxiety for patients with a high FCR score”. (lines 226-228)
Line 238: I see a discrepancy between the question you raised about satisfaction with cancer treatment and the specific interpretation you make regarding disruption to treatment or delays. If these aspects were not specifically asked in the question, it is difficult to know, which treatment aspects patients did consider. Maybe they were not satisfied with a physician or unfriendly staff etc. As it reads now, the causes for dissatisfaction cannot be identified.
Authors’ response: In fact, we assumed that the dissatisfaction addressed in the questionnaire comes from the measures put in place during the COVID-19 pandemic. The question in the questionnaire was: “Are you satisfied with the way your disease is currently managed?”. We implicitly assume that "currently" refers to “before the COVID-19 pandemic”. This hypothesis is further reinforced by the fact that this question is asked just before "are you satisfied with the measures put in place by the hospital" and "my illness is better/less well managed than before the COVID-19 pandemic".
Line 250-251: see my comments above. If you would compare prevalence rates of PTSD and anxiety of cancer patients before the pandemic you could make a stronger statement here.
Authors’ response: See our response above, line 200.
A systemic review and meta-analysis showed that the prevalence rate of anxiety was 10% in patients treated for cancer.
We now mention the prevalence of anxiety in cancer patients outside the epidemic context in the following sentence: In any case, the prevalence of anxiety assessed after the COVID-19 pandemic were much higher than the 10% reported in a previous systematic meta-analysis and a recent study, conducted before the epidemic context [26,27].
[26] Mitchell A, Chan M, Bhatti H, et al. Prevalence of depression, anxiety, and adjustment disorder in oncological, haematological, and palliative-care settings: a meta-analysis of 94 interview-based studies. Lancet Oncol. 2011 Feb;12(2):160-74. doi: 10.1016/S1470-2045(11)70002-X.
[27] Pitman A, Suleman S, Hyde N, Hodgkiss A. Depression and anxiety in patients with cancer. BMJ. 2018;361:k1415. doi: 10.1136/bmj.k1415
Line 252: Again I would tone down the implications. You did not explicitly measure additive stress (you measured positive lockdown experiences). And while fear of coming to the hospital due to the risk of a contamination is significant in both multivariate analyses, the lockdown experience is only significant for PTSD. Regarding anxiety, the strongest effect was found for fear of recurrence.
In order to tone down the implications, we deleted the term “additive in the sentence in line 249
We also modified the rest of the paragraph as follows: “Given that psychological distress in cancer patients has been associated with reduced treatment compliance, which may influence disease progression and mortality, any possible other cause of stress is of great concern for this population”. (line 254)
Line 264: If you want to make this argument you should more explicitly state the prevalence for fear of recurrence of 64.9% which is more than twice as high as the anxiety score of 30,5% for your study (without naming the exact numbers in the discussion).
Authors’ response: Thank you for this suggestion. We added the following sentence: « In our study, the prevalence of fear of cancer recurrence was more than twice as high as that of anxiety. » (line 262)
Line 269-270: Please avoid one sentence paragraphs. This information is a replication of past studies and is good to mention, since it strengthens the generalizability of your results. But, I would add some information on that and maybe other replicated regarding demographics or medical issues. That women are more affected also fits to the result that gynecological cancer is more relevant in anxiety than urological cancer.
Authors’ response: The single sentence in the text has been linked to the previous paragraph.
It is true that the medical context such as the type of cancer is a factor that can influence the psychological health of cancer patients. However, this was not the case in our study since the multivariate analysis was adjusted for cancer location. In order to give information to the reader, we therefore added the following sentence: “One should notice that the fact that women’s psychological health was more affected in our study was not related to the type of cancer as long as the multivariate analysis was adjusted for cancer location.” (line 267)
Line 286: here also a comparative prevalence rate of cancer patient’s mental health regarding anxiety and PTSD would help.
Authors’ response: The prevalence rate regarding anxiety was added as a response to previous reviewer’s comments.
(line 206): “In any case, the prevalence of anxiety assessed after the COVID-19 pandemic were much higher than the 10% reported in a previous systematic review and meta-analysis conducted before the epidemic context [26,27].”
Regarding PTSD, such comparative figures are very difficult to discuss in this part because PTSD outcome is linked to a particular traumatic event (accident, terrorist attack, natural disaster, pandemic for example) and is completely dependent on the event causing PTSD. It is therefore very difficult to compare them to other study contexts.
Reviewer 3 Report
In the discussion it would be interesting to add a comparison with available data of real worldpublished experiences
Author Response
REVIEWER 3
In the discussion it would be interesting to add a comparison with available data of real worldpublished experiences
Authors’ response: Thank you for this comment. To respond to the reviewer’s suggestion, we added the following sentence and references :
“In any case, the prevalence of anxiety assessed after the COVID-19 pandemic were much higher than the 10% reported in a previous systematic review and meta-analysis conducted before the epidemic context [26,27].” (line 206)
[26] Mitchell A, Chan M, Bhatti H, et al. Prevalence of depression, anxiety, and adjustment disorder in oncological, haematological, and palliative-care settings: a meta-analysis of 94 interview-based studies. Lancet Oncol. 2011 Feb;12(2):160-74. doi: 10.1016/S1470-2045(11)70002-X.
[27] Pitman A, Suleman S, Hyde N, Hodgkiss A. Depression and anxiety in patients with cancer. BMJ. 2018;361:k1415. doi: 10.1136/bmj.k1415
In addition, we added a sentence that refers to psychological health during the COVID-19 pandemic in the general population: “Previous studies conducted in the general population have shown the negative impact of the COVID-19 pandemic on psychological health, including fear of illness, fear of death, as well as the social effects of physical distancing [36,37]”. (line 247)
[36] Salari N, Hosseinian-Far A, Jalali R et al. Prevalence of stress, anxiety, depression among the general population during the COVID-19 pandemic: a systematic review and meta-analysis. Global Health 2020 Jul 6;16(1):57. doi: 10.1186/s12992-020-00589-w.
[37] Gray N, O’Connor C,Knowles et al. The Influence of the COVID-19 Pandemic on Mental Well-Being and Psychological Distress: Impact Upon a Single Country. Front. 2020 Nov 11;11:594115. doi: 10.3389/fpsyt.2020.594115.
Round 2
Reviewer 1 Report
the changes made are adequate for publication